# CD146 Delineates an Interfascicular Cell Sub-Population in Tendon That Is Recruited during Injury through Its Ligand Laminin-α4

**DOI:** 10.3390/ijms22189729

**Published:** 2021-09-08

**Authors:** Neil Marr, Richard Meeson, Elizabeth F. Kelly, Yongxiang Fang, Mandy J. Peffers, Andrew A. Pitsillides, Jayesh Dudhia, Chavaunne T. Thorpe

**Affiliations:** 1Comparative Biomedical Sciences, Royal Veterinary College, Royal College Street, London NW1 0TU, UK; nmarr@rvc.ac.uk (N.M.); apitsillides@rvc.ac.uk (A.A.P.); 2Clinical Sciences and Services, Royal Veterinary College, Hawkshead Lane, Hatfield AL9 7TA, UK; rmeeson@rvc.ac.uk (R.M.); efkelly88@gmail.com (E.F.K.); jdudhia@rvc.ac.uk (J.D.); 3Centre for Genomic Research, Institute of Integrative Biology, Biosciences Building, University of Liverpool, Crown Street, Liverpool L69 7ZB, UK; y.fang@liv.ac.uk; 4Institute of Ageing and Chronic Disease, University of Liverpool, Apex Building, 6 West Derby Street, Liverpool L7 9TX, UK; m.j.peffers@liv.ac.uk

**Keywords:** interfascicular matrix, tendon injury, CD146, LAMA4, Achilles tendon

## Abstract

The interfascicular matrix (IFM) binds tendon fascicles and contains a population of morphologically distinct cells. However, the role of IFM-localised cell populations in tendon repair remains to be determined. The basement membrane protein laminin-α4 also localises to the IFM. Laminin-α4 is a ligand for several cell surface receptors, including CD146, a marker of pericyte and progenitor cells. We used a needle injury model in the rat Achilles tendon to test the hypothesis that the IFM is a niche for CD146+ cells that are mobilised in response to tendon damage. We also aimed to establish how expression patterns of circulating non-coding RNAs alter with tendon injury and identify potential RNA-based markers of tendon disease. The results demonstrate the formation of a focal lesion at the injury site, which increased in size and cellularity for up to 21 days post injury. In healthy tendon, CD146+ cells localised to the IFM, compared with injury, where CD146+ cells migrated towards the lesion at days 4 and 7, and populated the lesion 21 days post injury. This was accompanied by increased laminin-α4, suggesting that laminin-α4 facilitates CD146+ cell recruitment at injury sites. We also identified a panel of circulating microRNAs that are dysregulated with tendon injury. We propose that the IFM cell niche mediates the intrinsic response to injury, whereby an injury stimulus induces CD146+ cell migration. Further work is required to fully characterise CD146+ subpopulations within the IFM and establish their precise roles during tendon healing.

## 1. Introduction

Tendons consist of highly aligned, collagen-rich fascicles bound together by a looser, less organised interfascicular matrix (IFM; also referred to as the endotenon). Our previous work established that the IFM plays an important mechanical role in tendon, allowing for sliding between fascicles and providing the tendon with additional extensibility [1,2,3]. These properties are provided by structural proteins, including elastin and lubricin [4,5,6,7]. The production and replacement of these components by IFM resident cells is likely critical to maintain tendon function. Our recent work demonstrates that protein turnover occurs more rapidly in the IFM compared to fascicles, indicative of greater metabolic activity [8,9]. IFM cell populations are, therefore, likely key contributors to maintaining tendon homeostasis and responding to injury. However, while it is evident that the IFM houses a population of cells that are morphologically distinct and present at a higher density compared to the elongated tenocytes resident within the fascicles [10], the roles of IFM-localised cell populations in tendon maintenance and repair remain largely undefined.

Several proteins characteristic of basement membranes have been shown to localise to the IFM [8,9,11]. Basement membranes are specialised matrices that interact with resident cells via cell surface receptors and are integral to tissue maintenance and repair, housing progenitor cell niches and providing structural integrity in musculoskeletal tissues [12,13]. Specifically, our previous work has demonstrated that the basement membrane protein laminin-α4 (LAMA4) localises predominantly to the IFM both in small and large animal models [11]. LAMA4 is a component of laminin-8, laminin-9, and laminin-14, and regulates basement membrane remodelling and function in cell niches in a variety of tissues, including adipose, endothelial, hematopoietic, muscular and neuronal tissues [14,15,16,17]. LAMA4 is also a ligand for several cell surface receptors, including CD146 (also known as MCAM/MUC18), and LAMA4-CD146 binding facilitates T-cell migration across basement membranes in response to inflammation [18]. CD146 is a marker of pericytes and progenitor cells, and previous studies using mouse models have demonstrated that CD146 is expressed by tendon-derived cells *in vitro*, and CD146+ cells are present at sites of tendon injury [19,20]. However, the location and distribution of these cells and their role in repair remain to be elucidated in both healthy and injured tendon.

A number of models have been developed for inducing tendon injury in rodents, which typically involve either complete or partial transection of the tendon, or injection of collagenolytic enzymes to degrade the tendon matrix [21,22,23]. While these models are useful for studying large tendon tears or traumatic injuries such as whole tendon rupture, they can produce variable results, particularly those that are enzyme-based, and induce damage to a large portion of the tendon [24,25]. Hence, there is a need for more precise models that induce localised damage that can be tracked over time. Puncture of the mouse Achilles tendon with a needle results in rupture of the central fibres, whilst the majority of the tendon structure remains undamaged, and this technique has previously been reported to have good reproducibility and be simpler and less severe than many other models of tendon injury [26,27]. However, previous studies have not been able to identify an IFM in the mouse Achilles [27], likely due to its small size, and therefore this model is not suitable for studying the role of IFM cell populations in tendon healing. By contrast, our recent work has established the presence of an IFM in the rat Achilles tendon [11], indicating that the rat may be a suitable model in which to study IFM cell populations in the context of tendon healing.

As well as providing the ability to study changes within a tissue in response to injury, *in vivo* models can also enable identification of circulating factors that alter as a result of injury and are therefore potential biomarkers of disease. It is increasingly recognised that circulating non-coding RNAs (ncRNAs) are promising biomarkers for a range of musculoskeletal diseases, including osteoarthritis and osteoporosis [28,29]. However, to the authors’ knowledge, only one previous study has attempted to establish if circulating ncRNAs can be used as biomarkers of tendon injury, demonstrating unique serum microRNA (miRNA) signatures in human rotator cuff tendinopathy and tears [30], and no studies have characterised circulating ncRNAs in animal models of tendon injury

The aim of this study was to optimise and characterise the needle injury model in the rat Achilles tendon and to use this model to test the hypothesis that the IFM acts as a tendon cell niche containing CD146+ cells that are recruited in response to tendon damage. We also aimed to establish how expression patterns of circulating ncRNAs alter with tendon injury and identify potential RNA-based markers of tendon disease.

## 2. Results

### 2.1. Injury Study

Puncture of the Achilles tendon with a needle was well tolerated by the animals, with minimal lameness and behavioural changes observed after surgery, and all animals returned to normal behaviour within 48 h. Pilot studies to optimise needle size revealed that wounds caused by 23G needles healed rapidly and were difficult to detect by histology 7 days post injury, whereas those caused by 19G needles resulted in damage towards the peripheral margins of the tendon as well as the central fibres and were less consistent between individuals (Appendix A). By contrast, wounds induced by 21G needles had a more consistent appearance between individuals and were easily identifiable on histological sections at all time points assessed.

21G needles were therefore used to induce injury in all subsequent experiments, resulting in the puncture of the central fibres of the tendon and formation of a lesion with clearly delineated borders, the size of which increased significantly throughout the study (Figure 1). Cell density within the lesion also increased significantly at day 21 compared to days 4 and 7, with infiltration of rounded cells apparent at all time points. There was a significant increase in histopathological scoring, as assessed by the Bonar score, in injured tendons compared to contralateral controls, but this did not differ between time points. It should be noted that increasing Bonar score for cellularity does not relate directly to the number of cells, as grade 3 indicates acellularity (see Appendix A), which likely accounts for the discrepancy between scoring results and cell density measurements between time points. There was no difference in Bonar score between control and sham-operated tendons, in which the tendon was isolated but no injury was induced, providing confidence that isolating the tendon does not affect the tendon itself. An overall weighted Kappa statistic of 0.9 was calculated, indicating excellent agreement between blinded scorers.

At day 4 post injury, acellular regions were present within the intact tendon immediately surrounding the lesion. By day 21, lesions were hypercellular, with chondroid changes, ascertained by observation of lacunae formation and ovoid cell morphology, visible in some areas surrounding the lesion, which were acellular at earlier time points. Polarising light microscopy revealed that collagen fibres immediately surrounding the lesion became disorganised, with no evidence of deposition of newly synthesised collagen at any time points. There was also significant disorganisation up to 500 µm proximal to the lesion site at all time points assessed, which appeared to worsen over time (Figure 1).

In contralateral control tendons, immunohistochemical analysis revealed that CD146 labelling was localised only to cells within the IFM and epitenon, and injury resulted in the recruitment of CD146+ cells to the vicinity of the lesion at days 4 and 7, and within the lesion at day 21 (Figure 2). Analysis of the distribution of CD146+ cells using 3D confocal imaging revealed an extensive interfascicular network of CD146 positivity within the uninjured tendon (Figure 3). At day 4 after injury, hypocellular regions were visible in the intact tissue surrounding the site of injury, with the majority of CD146+ cells detected in the epitenon. At 7 days after injury, CD146+ cells were present around the lesion, and by day 21, these cells were detected within the lesion itself (Figure 4 and Figure 5). These findings suggest that CD146+ cells are recruited and activated in the days following injury, but longer time frames are required for them to populate the lesion.

Labelling for the CD146 ligand laminin-α4 coincided with the distribution of CD146, localising to the IFM in uninjured tendons, with more widespread labelling within the lesion apparent at day 21 compared to days 4 and 7. The cell proliferation marker proliferating cell nuclear antigen (PCNA) was detected in the majority of cells within the IFM and fascicles in uninjured tendons, as well as within the lesion at all time points. Labelling for the tendon marker scleraxis (SCX) was minimal or absent in normal tendon and was restricted to a small number of cells in the IFM. In injured tendon, SCX+ cell numbers appeared to increase, although this was variable with minimal labelling in some, but not all lesions at early time points, and no detection at later time points. TGF-β, a growth factor that is upregulated in tendon injury, was detected throughout normal tendons, with greater staining observed within IFM cell populations compared to fascicular cells. At early time points, TGF-β+ cells were observed around but not within lesions, whereas at later time points labelling was also apparent within lesions, where it was both intra- and extra-cellular. There was minimal labelling of collagen type III, which was localised to the IFM in uninjured tendons. Collagen III was absent from the lesions in injured tendons at the earlier time points studied but was detectable at 21 days post injury (Figure 2).

CD146 and CD31 are both markers of perivascular cell lineages, with CD31 recognised as a late endothelial progenitor cell marker. To identify whether endothelial progenitors are potentially implicated in tendon healing dynamics, histological sections were labelled for CD31, with results revealing sparse staining of CD31 in the IFM in uninjured tendon, with little or no CD31 staining observed within the lesion at any of the time points studied, indicating little involvement of these cells in tendon healing (Figure 6).

To establish macrophage recruitment after injury, sections were labelled for the pan-macrophage marker CD14, pro-inflammatory macrophage marker CD86 and anti-inflammatory macrophage marker CD163. CD14 was detected in subsets of cells in both the IFM and fascicles in uninjured tendons, and around the lesion at early time points, with minimal expression within the lesion at day 21. CD86 was detected in a subset of both fascicular and interfascicular cells in normal tendon, with greater expression in the IFM. CD86+ cells were present in the lesion at all time points. By contrast, CD163 was detected in a small number of cells in the IFM in uninjured tendon and was only detected within the lesion at the later time points studied. These results indicate that there are differences in the timescales of the recruitment of pro- and anti-inflammatory macrophages, with the initial recruitment of pro-inflammatory macrophages being followed by the infiltration of anti-inflammatory macrophages at later time points (Figure 6).

To determine if circulating progenitor cells are recruited in response to injury, tendon sections were labelled for the bone-marrow-derived mesenchymal stem cell marker CD271, stromal-cell-derived factor-1 (SDF-1), a chemokine that recruits CXCR-4+ bone-marrow-derived cells and CXCR-4. CD271 was not detected in uninjured tendon. Minimal labelling was observed in injured tendon, with no detection in some samples at later time points. SDF-1 was also restricted to the IFM in uninjured tendons, with the detection of SDF-1, and CXCR-4-positive cells within the lesion at all time points studied (Figure 6). These results indicate that some of the sub-populations present within injury sites are recruited from extrinsic sources.

### 2.2. Serum RNA-seq

Analysis of small RNAseq data revealed significant regulation of several ncRNAs within serum as a result of injury. Most of these were miRNAs. A small number of differentially expressed (DE) transfer RNAs (tRNAs) (10) and other ncRNAs (27) were identified with *p* < 0.05. However, these were not subjected to further analysis as only one ncRNA, and no tRNAs had false discovery rate (FDR)-adjusted *p*-values < 0.1. Of the 16 DE miRNAs, 12 have previously been associated with connective tissue disorders, including arthritis, osteoporosis and cardiovascular disease (Table 1).

The results from the qRT-PCR validation of small RNAseq results are shown in Table 2. Due to limited serum RNA availability, it was only possible to compare samples at day 4 from sham and injured groups. For the majority of miRNAs, the differences observed between sham and injured samples were of a similar magnitude to those measured by next-generation sequencing. Unfortunately, due to insufficient serum, it was not possible to perform statistical analysis of the qRT-PCR results.

Ingenuity Pathway Analysis (IPA) of RNAseq data identified several molecular and cellular functions associated with DE miRNAs, including cell development (*p* = 0.046), growth and proliferation (*p* = 0.046), cell death (*p* = 0.044) and cellular movement (*p* = 0.037), as well as organismal injury and abnormalities (*p* = 0.048). According to the top scoring networks, shown in Appendix A, DE miRNAs between shams and injured tendons at day 4 were primarily associated with organismal injury and abnormalities, whereas those DE between injured tendons at day 4 and day 21 were associated with organismal injury and abnormalities, as well as cell movement, cell death and survival. Analysis of miRNA targets identified TGF-β (*p* < 0.0001) and TNF (*p* < 0.0001) as upstream regulators, which were associated with fibrosis (*p* < 0.0001), cell proliferation (*p* < 0.0001) and cell migration (*p* < 0.0001).

## 3. Discussion

In this study, we have optimised and characterised a simple, reproducible model of injury in the rat Achilles tendon. This model demonstrates that proteins expressed within the IFM in healthy tendon are present within needle-induced lesions, suggesting that IFM-localised cells migrate to populate sites of injury. However, the roles these cells play in tendon healing are yet to be determined.

The needle-induced model of tendon injury that we have used recapitulates many features of naturally occurring tendon injury, with disruption of the collagen fibres, alterations in cellularity, infiltration of inflammatory cells and degeneration over time, both within and remote from the lesion observed [47,48,49]. However, there are some limitations associated with the model chosen. Female rats were used to reduce the likelihood of rats fighting post surgery, and therefore differences in injury response between sexes could not be assessed. In addition, while the injury model is simple to perform and the acute nature of the injury means that profiling changes over time is straightforward, it does not completely recapitulate naturally occurring tendon injury in which microdamage accumulates over time. The surgical approach also results in damage to the structures surrounding the tendon during surgery. However, there were no observable changes to cells or ECM structure in sham-operated tendons, providing confidence that isolating the tendon prior to injury induction does not affect the tendon itself. Immunohistochemical approaches do not allow for the quantification of changes in protein abundance over time within the lesion, and the amount of available tissue limits the breadth of analysis that could be performed, such that it was not possible to measure changes in ncRNA signatures within the tendon tissue, which would have strengthened the study. It was not possible to measure lesion volume from 3D scans, and therefore lesion area was measured from histological sections. Whilst every effort was made to ensure comparable sections from the centre of each tendon were used, small differences in tendon orientation during sectioning may have affected the resulting measurements, which likely contributed to the variability in measured lesion area, particularly at later time points.

Analysis of protein expression patterns in uninjured tendons provided insights into the phenotype of IFM cell populations, which remain poorly characterised. As hypothesised, CD146+ cells localised to the IFM and the epitenon. Previous studies have shown that CD146-expressing cells are found in peri-vascular regions of the human Achilles tendon where they co-localise with the stem cell marker nestin [19], and the single-cell RNA sequencing of human tendon revealed three cell populations that expressed CD146, an endothelial population that is also CD31+, as well as two tenocyte populations [50]. In the current study, CD31 was also localised to the IFM, with fewer labelled cells than CD146, suggesting that there may be several CD146-expressing cell populations within the IFM, at least one of which co-expresses CD31, likely delineating endothelial progenitor cells [51]. In bone marrow, it has been demonstrated that CD146+/CD31− cells delineate mesenchymal progenitors [52]. Taken together, these results suggest that the IFM may act as a niche for several distinct progenitor cell populations. Distribution of LAMA4 was similar to CD146, supporting previous studies demonstrating localisation of LAMA4 to the IFM in both small and large animal models [11]. Anti-inflammatory macrophage marker labelling was restricted to the IFM in uninjured tendon, suggesting that IFM cell populations may have a role in the resolution of inflammation.

After injury induction, CD146+ cells were present around the lesion after 7 days, and within the lesion 21 days after injury. Similar labelling patterns of the CD146 ligand LAMA4 in both uninjured and injured tendon suggest that these cells were migrating from the IFM to populate the site of injury. This supports previous studies demonstrating increased laminin at tenotomy sites in murine flexor digitorum profundus tendons [53], and upregulation of the LAMA4 gene in tendinopathic compared to healthy human tendon [54]. Treatment of transected rat patellar tendons with connective tissue growth factor resulted in increased CD146+ in healing tissue at early but not late time points, which was associated with improved healing [55]. Interestingly, in the current study, the response of CD146+ cells were different to that observed in the study by Lee et al. [55] in which CD146+ cells appeared in healing tissue two days after injury, after which the numbers decreased. These differences are likely due to differences in the model used to induce injury, as well as structural and anatomical differences between the Achilles and patellar tendons. Previous studies have not been able to identify an IFM structure in the rat patellar tendon, which is significantly smaller than the Achilles tendon [56], and CD146+ cells are localised to the patellar tendon peritenon, rather than within the tendon itself [55,57], suggesting that cells recruited in response to patellar tendon injury may be extrinsic rather than intrinsic.

It has been demonstrated that CD146 promotes cell proliferation via mTORC2 signalling [58]. In other tissues, CD146 has a role in pathological angiogenesis during tumorigenesis [59], and blocking CD146 activity inhibits tumour growth [60]. CD146 also binds LAMA4 and allows for the extravasation of T-helper 17 cells, which promotes an inflammatory response in the central nervous system [18]. Similar patterns of CD146 and LAMA4 labelling were observed in the current study, suggesting that CD146 and LAMA4 binding may play a similar role in the response to tendon injury, although the dual labelling of injured tendons for these markers is required to confirm this. These data suggest a complex role of CD146+ cells in tissue healing, with additional studies required to elucidate the specific contribution of these cells to tendon healing and to determine if CD146/LAMA4 binding plays a role in cell response to tendon injury.

We selected a broad panel of markers to identify cells of tenogenic, perivascular, endothelial and immune origin, as well as ECM proteins associated with the IFM. Changes in labelling patterns of these markers with injury provide insight into the response of other cell populations to injury. In common with previous studies, both pro- and anti-inflammatory macrophage markers were identified in injured tendon [61,62]. SDF-1 and CXCR-4 were present at low levels within the lesion at all time points studied, and labelling for the bone marrow mesenchymal stem cell marker CD271 [63,64] was observed at day 4 and 7 but not day 21, supporting previous studies that show infiltration of circulating cells to injured tendon 24 h post injury, after which levels decline [65]. It is well established that CXCR-4 and SDF-1 are required for tissue regeneration [66], and delivering collagen scaffolds treated with SDF-1 improved healing in rat Achilles defect by increasing recruitment of CXCR4+ cells [67]; however, the origin of these CXCR4+ cells is yet to be determined. In addition, it should be noted that labelling for most of the cell markers studied was restricted to a small number of cells at day 7 post injury, indicating that there are subsets of cells present within the lesion at this time point that are yet to be identified.

While collagen type III was present within the IFM, there was a lack of collagen III labelling within the lesion, which is surprising, as this is considered a hallmark of tendon injury [68]. As response to injury was assessed over a relatively short period of time, it is possible that collagen III production would increase at later time points. However, deposition of type III collagen has been observed two weeks post injury in other studies in rat tendon [69,70].

Limited SCX expression was observed in either healthy or injured tendon. This fits with previous studies showing that while SCX is highly expressed in embryonic and postnatal tendon, its expression decreases with ageing in ovine tendon [71]. Furthermore, it has been described that SCX+ cells are activated in neonatal tendon injury and drive regenerative healing, while in in adult tendon, SCX+ cells are not recruited, and tendon healing occurs in a fibrotic manner, resulting in impaired function [72].

TGF-β showed similar labelling patterns to those reported previously, being predominantly localised to the IFM in normal tendon and increased staining being observed with injury [71,73]. Previous studies have implicated that TGF-β signalling has important roles in tendon development, homeostasis and injury, with activation of TGF-β in the IFM during development. This is predicted to result in the synthesis of major ECM proteins [74,75,76], and the dysregulation of TGF-β signalling is implicated in the ageing of the human Achilles tendon [77]. These results are supported by our study, with pathway analysis of DE serum miRNAs predicting upstream regulation by TGF-β signalling. In other tissues, TGF-β is involved in the regulation of angiogenesis in wound healing, along with CD146, laminins and other basement-membrane-associated proteins [78,79].

This is the first study to assess circulating ncRNAs in a surgically induced model of tendon injury. Whilst TGF-β was identified as a potential upstream regulator, no miRNAs that have previously been identified in tendon injury were differentially expressed, likely because miRNAs were assessed in serum, rather than within the injury site itself [30,80,81]. MiR-29a is the most well-studied miRNA in tendon. Studies have demonstrated that it is downregulated in tendon disease, and miR-29a therapy improves healing in experimentally induced tendon injury [81,82]. The miRNAs that were differentially expressed in serum were associated with connective tissue diseases or dysfunction and have been linked to a variety of connective tissue diseases and abnormalities, particularly arthritis [32,38,41]. Indeed, several of these miRNAs have been specifically identified in serum in previous studies [31,35,41]. While there is unlikely to be a single miRNA-based biomarker for tendon injury, previous studies have identified miRNA signatures that can distinguish between cancers [83]. Therefore future studies can build on the preliminary findings we report here to establish specific miRNA fingerprints as biomarkers of tendon disease.

In summary, we have characterised and optimised a simple needle injury model in the rat Achilles tendon, and we used this model to identify, for the first time, an IFM-localised CD146+ cell population that progressively accumulates at the site of injury. CD146 expression was accompanied by the upregulation of its ligand LAMA4, consistent with the IFM cell niche mediating the intrinsic response to injury. Use of this model will allow future studies to fully characterise CD146+ subpopulations within the IFM and their relative abundance at each repair phase post injury, as well as to establish how cell–ECM interactions prohibit or improve healing response and how IFM cell response can be modulated by therapeutics.

## 4. Materials and Methods

### 4.1. Animals

Female Wistar rats (*n* = 24, 12 weeks old, 276 ± 13 g; Charles River Company, Cambridge, UK) were randomised and housed in polypropylene cages in groups of 2, subjected to 12 h light/dark cycle with room temperature at 21 ± 2 °C and fed ad libitum with a maintenance diet (Special Diet Services, Chelmsford, UK). All procedures complied with the Animals (Scientific Procedures) Act 1986, were approved by the Royal Veterinary College Animal Welfare and Ethical Review Body (ID:2016-0096N; June 2017), were performed under project licence PB78F43EE and are reported according to the ARRIVE guidelines [84].

### 4.2. Tendon Needle Injury

Rats were anaesthetised (isoflurane; 2.5–3%), pre-operative analgesia was provided (0.05 mg/kg buprenorphine, sub-cutaneous), and their left hindlimb clipped and aseptically prepared (5% chlorhexidine solution, followed by 70% IMS). A custom-made foot stool was used to position the paw and allow the Achilles to be tensioned, and then the distal limb was draped with a fenestrated drape. A 6 mm midline longitudinal skin incision was made over the dorsal aspect of the left Achilles tendon, which was then isolated from the plantaris tendon and surrounding tissue with blunt dissection. A 21G hypodermic needle was used to puncture the tendon 2 mm proximal to the calcaneal insertion by passing the needle through the tendon once perpendicular to its surface [26,27]. The incision was closed with a single temporary suture (4–0 monocryl) to appose the skin edges and to move them away from the adjacent Achilles tendon as the tissue glue (INDERMIL^®^ Flexifuze™, Vygon, Swindon, UK) was applied, prior to removing the temporary suture. Sham procedures were performed (*n* = 2/time point), in which the skin was incised and the tendon was isolated but no injury was introduced. All surgeries were performed by one operator. Analgesia was provided for 48 h post operatively (0.05 mg/kg buprenorphine, sub-cutaneous). Post surgery, rats were returned to their pre-surgery groups, and behaviour was scored according to pre-defined criteria, assessing weight, appearance, lameness, unprovoked behaviour, body condition and respiration. In the 24 h post surgery, some rats displayed a small transient change in locomotory behaviour, but this returned to normal within 48 h.

Rats were euthanised 4, 7 and 21 days post surgery (*n* = 6/time point), blood samples were collected via cardiac puncture immediately after confirmation of death, and Achilles tendons were harvested within 2 h post mortem. The right Achilles tendon of each animal acted as an uninjured control. Tendons were embedded in optimal cutting temperature embedding matrix (OCT; Cell Path, Newtown, UK) and snap frozen in hexane cooled on dry ice (*n* = 4/time point) and stored at −80 °C, or fixed in 4% paraformaldehyde (PFA; *n* = 2/time point) overnight at 4 °C, then stored in Dulbecco’s phosphate-buffered saline (calcium and magnesium free) with 0.05% sodium azide to preserve the antigens prior to analysis. Blood samples were centrifuged at 1500 RCF for 10 min., and serum was collected and stored at −80 °C prior to analysis.

Pilot studies were performed to optimise time points and needle size, comparing 19, 21 and 23G needles and injury response at days 1, 4, 7 and 21 post injury (*n* = 2 per gauge size and time point). A 21G needle produced the most consistent results, and this needle size was therefore used in all subsequent studies.

### 4.3. 2D Analysis of Cell Response to Injury

OCT-embedded tendons were serially sectioned coronally at a thickness of 12 µm and dried at room temperature (RT) for 2 h to allow sections to adhere to slides, before storage at −80 °C. Slides were thawed, fixed in ice cold methanol:acetone (1:1; 5 min.), washed in dH_2_O and stained with haematoxylin and eosin using standard protocols (*n* = 6 sections per sample; 4 samples per time point; Gemini AS Automated Slide Stainer, Thermo Scientific, Waltham, MA, USA). Sections were imaged using a slide scanner and ×20 objective (Zeiss Axioscan, Oberkochen, Germany) and additionally using polarising light microscopy (DMRA2 upright microscope; objective: 20× HC PL FLUOTAR PH2; DFC550 colour camera; LAS-X version 3.7 software (Leica Microsystems, Wetzlar, Germany)) to allow for the assessment of collagen organisation. Resulting images from injured, sham and contralateral control tendons were scored by two blinded scorers, using a modified Bonar scoring system as described by Fearon et al. [47] to assess cellularity, cell morphology, tissue organisation and collagen alignment. Grading categories are shown in Appendix A. Inter-observer variability was assessed by calculating linear weighted Kappa statistics [85] using an online software tool (https://vassarstats.net/kappa.html, accessed on 15 December 2019). In addition, lesion area and cellularity were calculated from images of the 2 adjacent sections obtained from the central region of the tendon using QuPath software [86]. The lesion area was outlined manually, and area and cells/mm^2^ were calculated.

### 4.4. Statistical Analysis

A D’Agostino and Pearson test was used to determine if the data followed a normal distribution. Nested one-way ANOVA followed by Sidak’s multiple comparisons test was used to determine differences in histopathological scores, lesion size and cellularity between control and injured groups across time points.

### 4.5. Immunolabelling

Following methanol:acetone fixation, sections (*n* = 4 per sample, 4 samples per time point) were washed with tris-buffered saline (TBS) and blocked with 10% serum in TBS with 1% bovine serum albumin (BSA) for 2 h at RT. Primary antibody incubations were performed overnight at 4 °C (see Appendix A for blocking conditions, antibody details and dilutions). Following TBS washes, sections were treated with 0.3% hydrogen peroxide for 15 min., and secondary antibodies were applied for 1 h at RT. Staining was developed with 3,3’-diaminobenzidine (5 min., Vector Labs, San Francisco, CA, USA), and sections were counterstained with haematoxylin for 30 s, dehydrated (Gemini AS Automated Slide Stainer, Thermo Scientific) and mounted with DPX. Sections were cured overnight and imaged using brightfield microscopy (DM4000B upright microscope; objectives: 10× HC PL FLUOTAR PH1, 20× HC PL FLUOTAR PH2; DFC550 colour camera; LAS-X version 3.7 software (Leica Microsystems)).

### 4.6. 3D Analysis of Cell Response to Injury

To enable cell response to injury to be visualised in 3D, PFA-fixed tendons (*n* = 2 per time point) were immunolabelled and optically cleared using HISTO™ solutions (Visikol, Hampton, NJ, USA) according to our recently developed protocol [11]. Briefly, following permeabilisation and blocking, tendons were incubated with primary antibodies for CD146 (1:100; Abcam: ab75769, Cambridge, UK) for 96 h to allow for the complete diffusion of the antibodies through the full thickness of the tendon. Secondary antibody incubation (Alexa Fluor^®^ 594 Goat anti-rabbit IgG, Invitrogen, 1:500, Waltham, MA, USA) was performed for 96 h, followed by overnight nuclei counterstaining with DAPI. Some tendons were additionally stained with Col-F (20 µm; ImmunoChemistry Technologies, Bloomington, MN, USA), a fluorescent probe that binds collagen and elastin, during the nuclei counterstaining step to allow for the visualisation of collagen within the tendon in 3D [87]. Tendons were then washed, dehydrated in methanol and cleared by incubation in HISTO™-1 and -2 solutions for at least 72 h. Tendons were stored in HISTO™-2 prior to imaging.

### 4.7. Confocal Imaging

Tendons were immersed in HISTO™-2 on a glass-bottom dish for imaging. Serial optical sections (z step size: 2.5 µm) were acquired throughout the depth of the tendon using a Leica TCS SP8 laser scanning confocal microscope equipped with a motorised stage. Images were acquired with a HC PL FLUOTAR 10×/0.32 dry objective lens at a resolution of 1024 × 1024 px, a pinhole size of 1 Airy unit, a frame average of 1 and a line average of 2. Tile scans of tendons were captured using light-emitting lasers at 405 (blue channel; DAPI) and 561 (red channel; Alexa Fluor^®^ 594) nm to detect fluorescent signal, with low laser power (<10%), and scanning speed set to 600  Hz. In injured tendons, the site of needle injury was identified based on both macroscopic appearance, with thickening observed around the injury site, and microscopic detection of both hypo- and hyper-cellular areas in this region. 3D rendering and projections were performed and visualised using Leica LAS X software (version 3.5.5) within the 3D module.

### 4.8. RNA Isolation, cDNA Library Preparation, and Small RNA Sequencing (RNA-seq)

RNA was extracted from 100–200 µL serum collected from rats in the following groups using miRNeasy kits (Qiagen, Crawley, UK) according to the manufacturer’s instructions: day 4 sham, day 4 injured, day 21 injured (*n* = 3/group). RNA quantity was determined by Qubit assay, and quality was assessed using an RNA high-sensitivity bioanalyser chip (Agilent, Stockport, UK). The NEBNext Small RNA Library Prep Set for Illumina was used for library preparation (New England Biolabs, Ipswich, MA, USA). Samples were amplified for 35 cycles and libraries were sequenced on an Illumina NovaSeq 6000 (Illumina, San Diego, CA, USA) to generate 2 × 150 bp paired-end reads. Resulting data are deposited on NCBI GEO, accession; E-MTAB-9509.

### 4.9. Data Processing

Sequence data were firstly processed, including basecalling and de-multiplexing of indexed reads, using CASAVA version 1.8.2, and adapter and quality trimming, using Cutadapt version 1.2.1 [88] and Sickle version 1.200, to obtain fastq files. The quantification of ncRNAs in the libraries was performed using Salmon transcripts quantification tool version 0.14.2 [89] against an index created using Salmon index tool with optional parameter -k 15 on combined reference sequences, which was sourced from Ensembl rat ncRNA data (ftp://ftp.ensembl.org/pub/release-98/fasta/rattus_norvegicus/ncrna/Rattus_norvegicus.Rnor_6.0.ncrna.fa.gz, accessed on 26 August 2020), UCSC rat tRNA data (http://gtrnadb.ucsc.edu/GtRNAdb2/genomes/eukaryota/Rnorv6/rn6-mature-tRNAs.fa, accessed on 26 August 2020) and miRbase rat mature miRNA data (ftp://mirbase.org/pub/mirbase/CURRENT/genomes/rno.gff3, accessed on 26 August 2020). Before merging the sequences from the three sources, the miRNA sequences in Ensembl rat ncRNA data, and duplicated sequences in tRNA UCSC rat data were excluded.

DE analysis was performed in R environment using the package DESeq2 [90]. Following a split of the ncRNA count data into 3 sub-datasets—the miRNA dataset, tRNA dataset and other ncRNA datasets—the DE analysis was applied to each sub-dataset, respectively. Briefly, assessment of data variation and detection of outliers was performed through comparing variations of within and between sample groups using principle component analysis (PCA) and correlation analysis; handling library size variation using DESeq2 default method; formulating data variation using negative binomial distributions; modelling data using a generalised linear model; computing log2 Fold Change (logFC) values for required contrasts based on model fitting results through contrast fitting approach, evaluating the significance of estimated logFC values by Wald tests; adjusting the effects of multiple tests using false discovery rate (FDR) approach to obtain FDR-adjusted P-values [91]; and defining significantly DE ncRNAs as those with an FDR-adjusted *p*-value  < 10%.

### 4.10. Pathway Analysis

Networks and functional analyses were generated using Ingenuity Pathway Analysis (IPA; Qiagen, Crawley, UK) on the list of differentially abundant genes with *p*  <  0.05. For network generation, datasets containing miRNA identifiers for DE miRNAs between shams and injured tendon at day 4, and injured tendons at day 4 and day 21, and the corresponding fold change in expression were uploaded into the application. These were analysed using the core analysis module, and networks were algorithmically generated based on connectivity. The functional analysis identified the biological functions and diseases most significant to the dataset. A right-tailed Fisher’s exact test was used to calculate *p*-values.

Target filter in IPA was used to identify targets of the miRNAs that were DE. Datasets were filtered for relevant cell type (fibroblasts), disease states (connective tissue disorders, inflammatory disease and response, organismal injury and abnormalities, skeletal and muscle disorders), and confidence (experimentally observed and high (predicted)). Identified targets were reanalysed in IPA to identify potential regulators.

### 4.11. qRT-PCR Validation

A panel of miRNAs were selected from those that showed the greatest DE in small RNA-seq data with FDR-adjusted *p*-values < 0.1. Control miRNAs were selected according to Zhou et al. [92], ensuring < 0.1-fold change between experimental groups. rno-let-7c-5p has previously been reported as the most stable miR in extracellular vesicles [93], and rno-miR-140-3p has been reported as a control miR in serum [94]. miR-103a-3p was used as an additional control, as recommended by Qiagen. Details regarding selected miRNAs, corresponding sequences and product codes are shown in Appendix A. A miRCURY LNA RT Kit (Qiagen, Crawley, UK) was used to reverse transcribe 32 ng of RNA to cDNA according to the manufacturer’s instructions. Relative expression of miRNAs was assessed using miRCURY LNA SYBR Green PCR kit and appropriate miRCURY LNA miRNA PCR assay primers (Qiagen, Crawley, UK) following the manufacturer’s instructions. Thermal cycling was performed on a CFX96™ Connect Real Time PCR detection system (Bio-Rad, Watford, UK). The cycling conditions comprised initial heat activation at 95 °C for 2 min, 40 cycles at 95 °C for 10 s and 56 °C for 60 s followed by a melt curve analysis. qRT-PCR reactions were performed in triplicate and expression was normalised to the reference genes and control samples using the 2^−ΔΔCT^ method [95]. Comparisons were made between serums from sham and injured animals at day 4 due to a lack of RNA remaining from the day 21 injured samples.

## Figures and Tables

**Figure 1 ijms-22-09729-f001:**
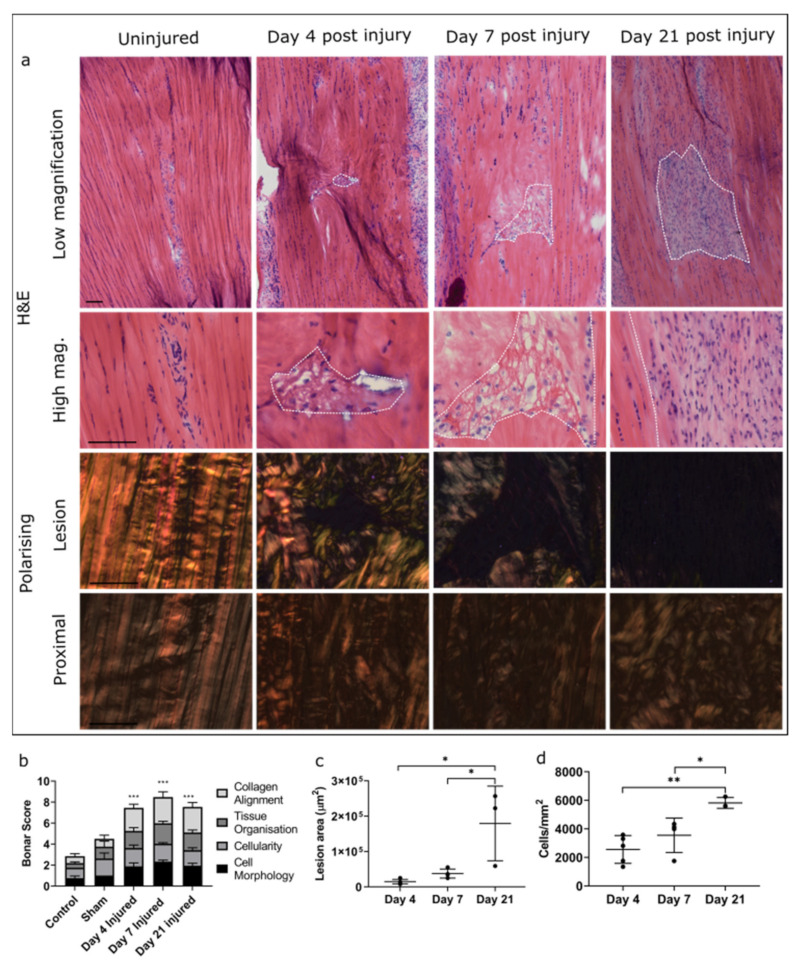
Effect of injury on tendon morphology and cellularity. H&E stained tendon sections visualised under bright and polarising microscopy demonstrate alterations in lesion appearance over time in injured tendons compared to contralateral controls. Lesions are outlined by dotted lines. Alterations in collagen organisation were also observed up to 500 µm proximal to the lesion. Scale bar = 100 µm (**a**). Scoring was based on the Bonar scoring system and demonstrated a significant increase in overall score in injured compared to control and sham samples (indicated by ***), but no difference between injury time points (**b**). Lesion area (**c**) and cellularity (**d**) were significantly greater at day 21 than at day 4 or day 7. Data are displayed as mean ± S.D; *n* = 4 per time point. Significance indicated by *: * = *p* < 0.05, ** = *p* < 0.01, *** = *p* < 0.001.

**Figure 2 ijms-22-09729-f002:**
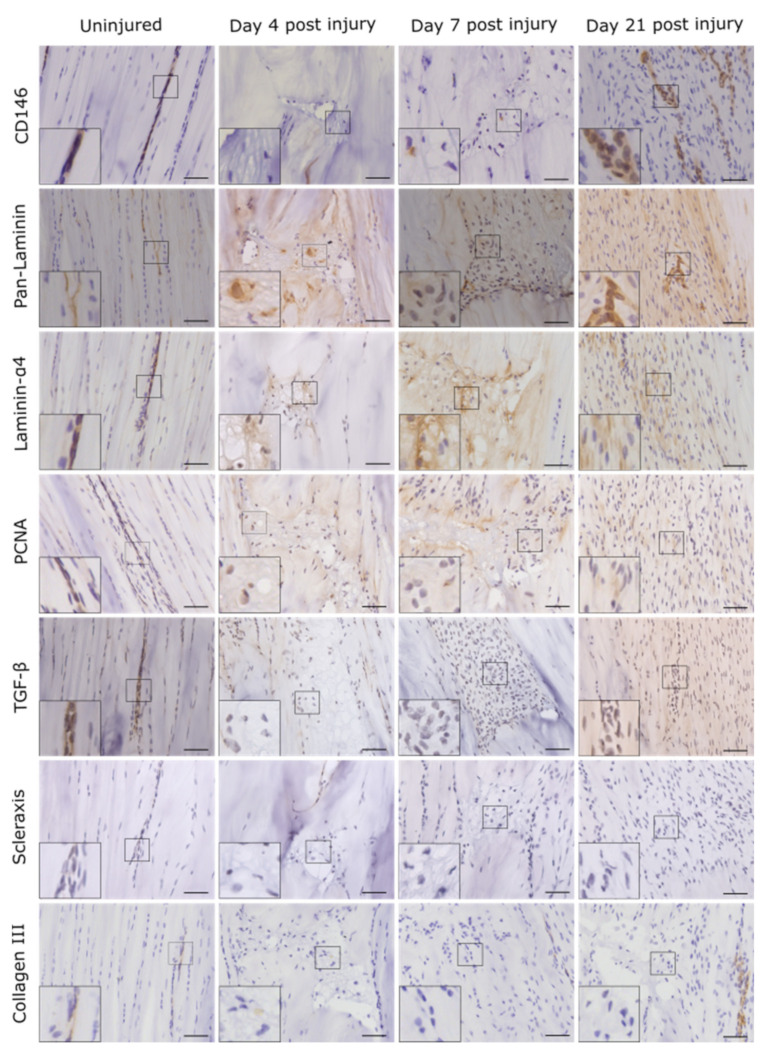
Injury response over time was assessed using immunohistochemistry for a panel of markers. Images show the lesion site in injured tendons and the corresponding region in uninjured tendons. Positive staining is brown, with cell nuclei counterstained blue. Inset shows the region of interest magnified 2.5 times. Scale bar = 50 µm.

**Figure 3 ijms-22-09729-f003:**
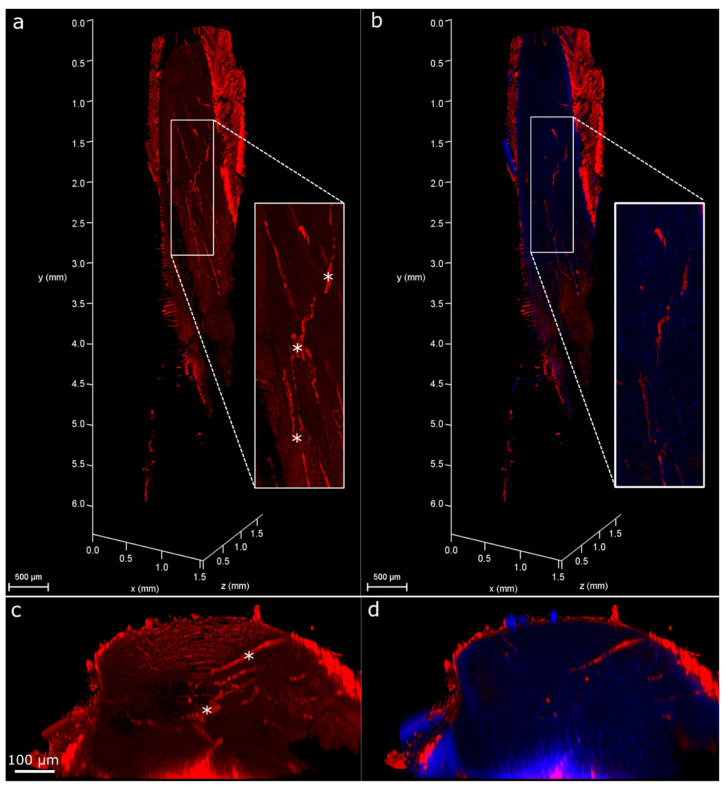
Three-dimensional visualisation of uninjured tendons labelled for CD146. Reconstruction of Achilles tendon labelled for CD146 (red, **a**,**b**) and with nuclei counterstained (blue, **b**). Reconstructions have been clipped longitudinally to allow for visualisation of CD146 within the tendon core. Inset shows magnified region. Transverse (xz) views of reconstructions (**c**,**d**) show a complex network of CD146 in uninjured tendon, localised to the IFM (indicated by *) and epitenon.

**Figure 4 ijms-22-09729-f004:**
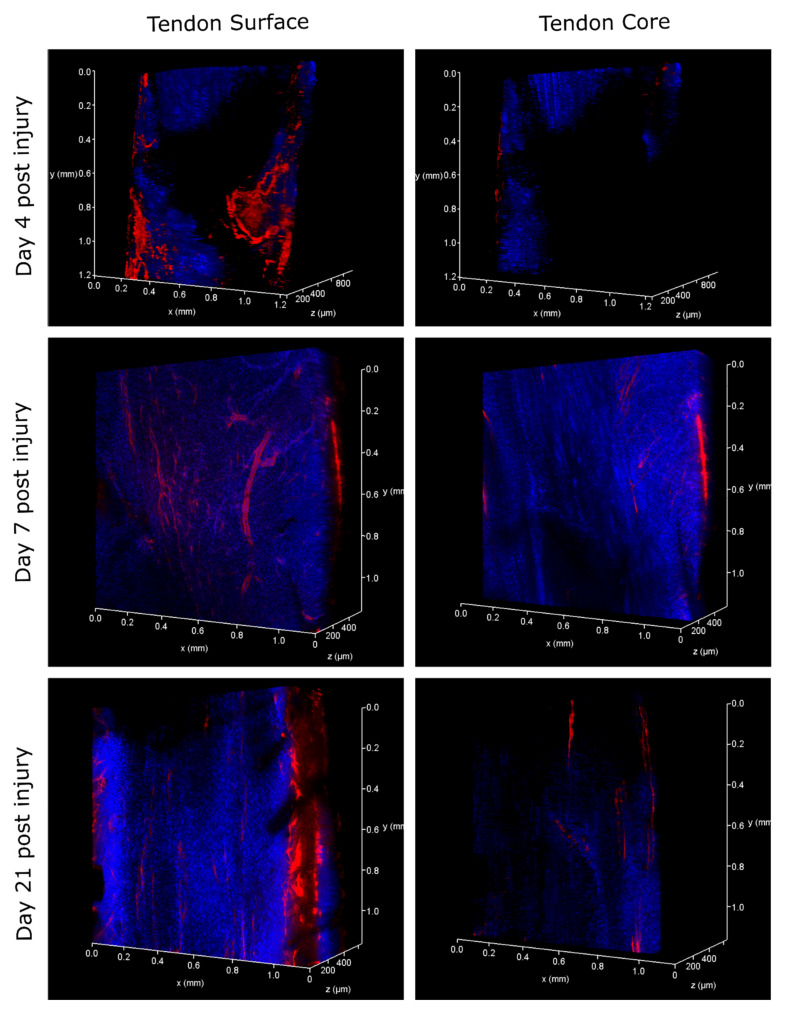
Three-dimensional visualisation of lesion sites in injured tendons at days 4, 7 and 21 after injury. CD146 is labelled red, with nuclei counterstained blue. At day 4 post injury, the area surrounding the lesion was mainly acellular, with CD146 expression restricted to the epitenon. By day 7, some CD146 positive-labelling could be visualised surrounding the lesion, and by day 21 CD146 expression was present within the lesion site.

**Figure 5 ijms-22-09729-f005:**
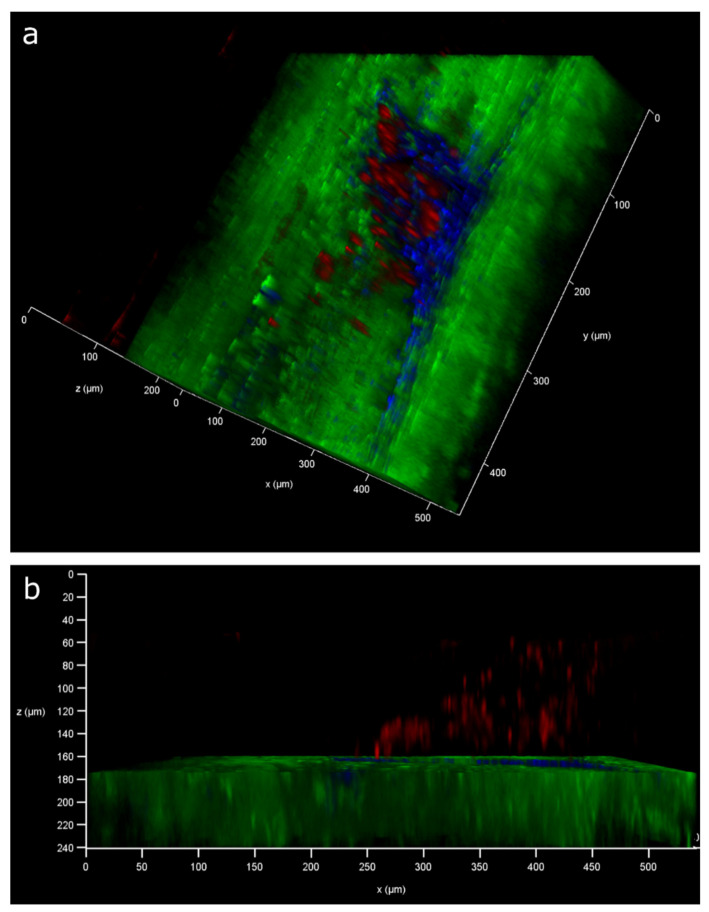
Three-dimensional visualisation of lesion site 21 days after injury. CD146 is labelled red, collagen is labelled green and nuclei labelled blue. Blue and green channels have been digitally removed in the centre of the tendon to allow for the visualisation of CD146-positive labelling within the lesion in sagittal (**a**) and transverse (**b**) views.

**Figure 6 ijms-22-09729-f006:**
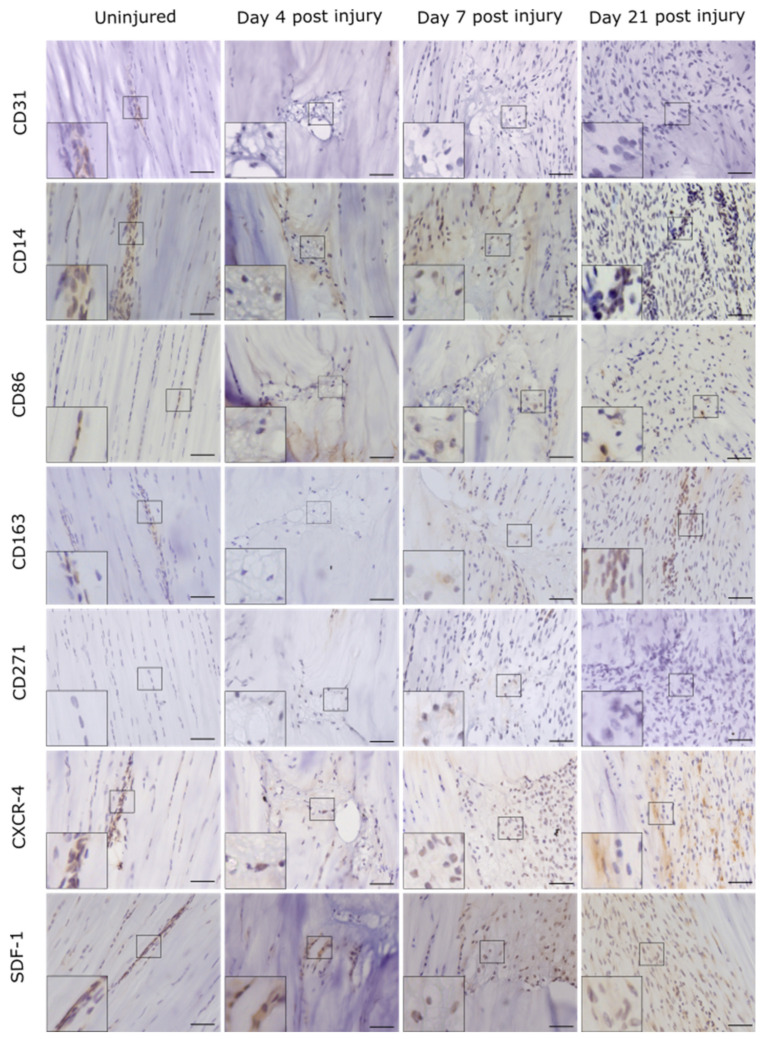
Injury response over time was assessed using immunohistochemistry for a panel of markers for progenitor cells, pericytes and macrophages. Images show the lesion site in injured tendons and the corresponding region in uninjured tendons. Positive staining is brown, with cell nuclei counterstained blue. Inset shows the region of interest magnified 2.5 times. Scale bar = 50 µm.

**Table 1 ijms-22-09729-t001:** Serum miRNAs differentially expressed between sham and injured tendons at day 4, and injured tendons at day 4 and day 21. *p*-values < 0.05 and FDR-adjusted *p*-values < 0.1 are displayed in bold.

	Log_2_ Fold Change		*p*-Value		FDR-adj. *p*-Value		
ID	D4 Sham v. Injured	Injured D4 v. D21	D4 Sham v. Injured	Injured D4 v. D21	D4 Sham vs. Injured	Injured D4 vs. D21	Disease Associations
miR-365-5p	2.21	0.10	**0.00**	0.88	**0.08**	1.00	Osteoporosis [31]
miR-338-5p	1.97	0.16	**0.00**	0.80	**0.10**	1.00	Rheumatoid arthritis [32]
miR-143-3p	1.63	1.10	**0.00**	**0.02**	**0.08**	0.79	Rheumatoid arthritis [33]
miR-145-3p	1.55	1.08	**0.01**	0.05	0.20	1.00	Heart disease [34]
miR-195-3p	1.52	0.00	**0.00**	1.00	**0.08**	1.00	Heart disease [35]
miR-676	1.16	0.32	**0.00**	0.43	0.17	1.00	Gastric cancer [36]
miR-10a-5p	1.10	0.86	**0.01**	**0.04**	0.30	1.00	Osteoarthritis [37]
miR-195-5p	0.78	1.76	0.25	**0.01**	1.00	0.67	Osteoarthritis [38]
miR-199a-5p	0.74	1.29	0.24	**0.04**	1.00	1.00	Lung fibrosis [39]
miR-145-5p	0.71	1.80	0.36	**0.02**	1.00	0.79	Rheumatoid arthritis [40]
miR-150-5p	−0.87	−0.26	**0.01**	0.44	0.35	1.00	Rheumatoid arthritis [41]
miR-451-5p	−1.06	−0.37	**0.01**	0.39	0.36	1.00	Diabetic neuropathy [42]
miR-200a-3p	−1.37	0.55	**0.00**	0.13	**0.08**	1.00	Lung cancer [43]
miR-672-5p	−1.80	1.63	**0.00**	**0.00**	**0.09**	0.30	Osteonecrosis [44]
miR-218a-5p	−2.09	0.72	**0.00**	0.24	**0.08**	1.00	Hypertension [45]
miR-504	−3.38	1.11	**0.00**	0.16	**0.09**	1.00	Lung cancer [46]

**Table 2 ijms-22-09729-t002:** qRT-PCR validation of serum miRNAs differentially expressed between sham and injured tendons at day 4.

ID	Gene ID	Log_2_ Fold Change: Sham vs. Injured (Day 4)
miR-365-5p	MIMAT0017184	1.49
miR-338-5p	MIMAT0004646	1.23
miR-143-3p	MIMAT0000849	2.79
miR-195-3p	MIMAT0017149	0.37
miR-672-5p	MIMAT0005327	−4.06
miR-218a-5p	MIMAT0000888_1	−2.17

## Data Availability

RNA-seq data are deposited on NCBI GEO, accession; E-MTAB-9509. All datasets generated and analyzed during the current study are available from the corresponding authors on reasonable request.

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
