# Peer review of "CD146 Delineates an Interfascicular Cell Sub-Population in Tendon That Is Recruited during Injury through Its Ligand Laminin-α4"

_ijms, 2021, doi:10.3390/ijms22189729_

Round 1

Reviewer 1 Report

In their study, the authors properly describe the setup of a needle punch injury model in the rat Achilles tendon and show a comprehensive characterization of the established injury model by immunohistochemical and structural analyses, clearly showing the recruitment of CD146+ cells from the endotenon to the lesion site over time.

Major concern:

The rationale of miRNA analyses in context with CD146 in tendon injury is not clear, moreover these part seems disconnected from the main study. Therefore, I would recommend to remove the miRNA part since the manuscripts is lacking deeper evaluation of miRNA data (no statistics etc). Additionally, title and abstract are missing information on miRNA related study content.

The authors state that CD246 positive cells are recruited through its ligand laminin a4. This statement should be proofed by double staining of CD146 and LAMA4 in the respective tissue samples.

Minor points:

Figure 1:

Please explain the “cell type” in HE stained section of day 7 p.i.

Bonar Score: How does it come that there are no differences determined in cellularity between the e time points? In histological sections cellularity seems rather different.

Figure 2:

Inserts sometimes are out of focus and scale bars are missing.

It would be very informative to see a collagen type I staining in the injured tendons.

Figure 6:

Inserts sometimes are out offocus and scale bars are missing.

Additional files:

Figure A1:

Why are the images of 23g day4 and day21 and 19g day1 and day 7 not provided, please explain. 

Author Response

Reviewer 1

In their study, the authors properly describe the setup of a needle punch injury model in the rat Achilles tendon and show a comprehensive characterization of the established injury model by immunohistochemical and structural analyses, clearly showing the recruitment of CD146+ cells from the endotenon to the lesion site over time.

Major concern:

The rationale of miRNA analyses in context with CD146 in tendon injury is not clear, moreover these part seems disconnected from the main study. Therefore, I would recommend to remove the miRNA part since the manuscripts is lacking deeper evaluation of miRNA data (no statistics etc). Additionally, title and abstract are missing information on miRNA related study content.

Response: In line with the reviewer’s comments, we have included more details on the rationale of assessing serum miRNA levels in tendon injury (ln83-87). To our knowledge, this is the first study to report changes in circulating miRs in an induced model of tendon injury. While looking at miRNAs within injured tendon would have strengthened the study, unfortunately we did not have sufficient tissue to conduct these experiments, this information is included in the discussion (ln268-269). While we did not have sufficient serum to perform statistical analysis of the qRT-PCR validation experiments, we were able to perform statistical analysis of the RNAseq data and we have clarified this in the manuscript (ln218&233). We have also added information regarding the serum miRNA analyses in the abstract (ln20-22&27-28).

The authors state that CD246 positive cells are recruited through its ligand laminin a4. This statement should be proofed by double staining of CD146 and LAMA4 in the respective tissue samples.

Response: We agree that double labelling of CD146 and LAMA4 is required to support this point. Unfortunately, as both CD146 and LAMA4 antibodies are raised in rabbit, we have been unable to perform dual labelling, despite extensive optimisation of techniques. We have also been unable to find suitable antibodies raised in other species that react with these proteins in the rat with high enough specificity. We have added this point to the discussion (ln 315-6).

Minor points:

Figure 1:

Please explain the “cell type” in HE stained section of day 7 p.i.

Response: There are likely several cell types that infiltrate the lesion by day 7, however we cannot confidently identify these from H&E imaging. Indeed labelling for the majority of cell markers was low at day 7, indicating there are several populations that we have not identified. We have further commented on this in the discussion (ln331-334).

Bonar Score: How does it come that there are no differences determined in cellularity between the e time points? In histological sections cellularity seems rather different.

Response: We thank the reviewer for raising this interesting point. Indeed, directly measuring cellularity within the lesion showed an increase in cells/mm2 at day 21 compared to days 4 and 7. However, increasing Bonar score for cellularity does not simply mean increasing cell number (see grading system below); as grade 3 indicates acellularity. We have clarified this (ln120-123) and also added the scoring system as supplementary information (Table A1).

Grade

0

1

2

3

Cell morphology

Inconspicuous elongated spindle shaped nuclei with no obvious cytoplasm at light microscopy

Increased roundness: nucleus becomes more ovoid to round in shape without conspicuous cytoplasm

Increased roundness and size; the nucleus is round, slightly enlarged and a small amount of cytoplasm is visible

Nucleus is round, large with abundant cytoplasm and lacuna formation (chondroid change)

Cellularity

Mainly discrete cells

Hyper cellular, in rows and/or increased cell numbers

Areas of hypo as well as hyper cellularity

Area of assessment is mostly a-cellular

Vascularity

No vessels present in FOV/ Inconspicuous blood vessels coursing between bundles

Occasional cluster of vessel

2–3 clusters of capillaries/vessels

Areas with greater than 3 clusters

Tissue Organisation

Collagen fibres arranged linearly with cell nuclei aligned with long axis of tendon

Small loss of collagen and cell alignment in regions

Moderate loss of collagen and cell alignment

Complete loss of collagen and cell alignment, matrix appears disorganised

Figure 2:

Inserts sometimes are out of focus and scale bars are missing.

Response: We have reviewed Fig 2 and replaced images which were out of focus. All images are shown at the same magnification, for clarity we have added scale bars to all images.

It would be very informative to see a collagen type I staining in the injured tendons.

Response: We agree that labelling for type 1 collagen would add valuable information to the study; unfortunately we do not have any tendon sections remaining to perform this analysis.

Figure 6:

Inserts sometimes are out offocus and scale bars are missing.

Response: We have reviewed Fig 6 and replaced images which were out of focus. All images are shown at the same magnification, for clarity we have added scale bars to all images.

Additional files:

Figure A1:

Why are the images of 23g day4 and day21 and 19g day1 and day 7 not provided, please explain. 

Response: We performed two optimisation experiments, in the first of which we assessed response at days 1 and 7 post injury using 23g and 21g needles. Based on these results, we observed very low cellularity in the lesion in day 1 samples, and found the lesion site difficult to identify in tendons injured with 23g needles. Therefore these time points and gauges were not used in the second optimisation experiment to reduce animal numbers. In the second optimisation experiment we added a larger needle size (19g) and an intermediate and longer time point (4 & 21 days), comparing these to day 4 and 7 using 21g needles. We have clarified this in the figure legend (Fig. A1). 

Reviewer 2 Report

The major problems come from the confusing presentation of those figure data. The main data of this manuscript is consisting of many figures and when the authors put them together, they should make sure the figures can tell the story clearly and easily. However, I think in most figures, the authors did not point out the injury zone/area in each image, which made them confusing to understand. For example, in figure 1,2, 6, the authors did not tell direct message to show day 4-day 21 were samples of injury. No idea where was the injury site. In figure 4, where was the lesion site?

Minor issue:

1, when  tendons were incubated with primary and secondary antibodies, the incubation time was 96 h. Why was it so long?

2, "s those caused by 19G needles resulted in damage towards the peripheral margins of the tendon as well as the central fibres and were less consistent between individuals (Figure A1)." I did not see any data related to what the authors talked about in figure A1.

Author Response

Reviewer 2

The major problems come from the confusing presentation of those figure data. The main data of this manuscript is consisting of many figures and when the authors put them together, they should make sure the figures can tell the story clearly and easily. However, I think in most figures, the authors did not point out the injury zone/area in each image, which made them confusing to understand. For example, in figure 1,2, 6, the authors did not tell direct message to show day 4-day 21 were samples of injury. No idea where was the injury site. In figure 4, where was the lesion site?

Response: In line with the reviewer’s suggestions, in figure 1 we have outlined the injury site, and clarified in the figures that day 4,7 and 21 refers to time points post-injury. We did try outlining the lesion in figures 2 and 6, but due to the relatively small size of the panels this made the images difficult to interpret. We have added to the figure legend to clarify that the lesion site is shown in the injured tendons. Unfortunately, it is difficult to identify the precise lesion site in figure 4 as collagen was not labelled in samples from earlier time points. However, we are able to identify the region in which the injury was created based on both macroscopic and microscopic appearance and we have clarified this in the methods (ln483-486).

Minor issue:

1, when  tendons were incubated with primary and secondary antibodies, the incubation time was 96 h. Why was it so long?

Response: We have previously shown that extended incubation times are required to allow antibodies to diffuse through the entire depth of the rat Achilles tendon. We have now clarified this point in the methods (ln466-467). For IHC labelling of tendon sections, overnight incubations were performed for primary, and 2h incubations for secondary antibodies, as stated in the methods (ln 452-455)

2, "s those caused by 19G needles resulted in damage towards the peripheral margins of the tendon as well as the central fibres and were less consistent between individuals (Figure A1)." I did not see any data related to what the authors talked about in figure A1.

Response: We have now included a magnified inset showing damage to the periphery of a tendon injured with 19g needle in figure A1